# CAN FAIR FEDERATED LEARNING REDUCE THE NEED FOR PERSONALIZATION?

## ABSTRACT

Federated Learning (FL) allows edge devices to collaboratively train machine learning models without sharing data. Since the data distribution varies across clients, the performance of the federated model on local data also varies. To solve this, *fair* FL approaches attempt to reduce the accuracy disparity between local partitions by focusing on clients with larger losses; while *local adaptation* personalizes the federated model by re-training it on local data—providing a device participation incentive when a federated model underperforms *relatively* to one trained locally. This paper evaluates two Fair Federated Learning (FFL) algorithms in this relative domain and determines if they provide a better starting point for personalization or supplant it. Contrary to expectation, FFL does not reduce the number of underperforming clients in a language task while doubling them in an image recognition task. Furthermore, fairness levels which maintain performance provide no benefit to relative accuracy in federated or adapted models. We postulate that FFL is unsuitable for our goal since clients with highly accurate local models require the federated one to have a disproportionate local accuracy to receive benefits. Instead, we propose Personalization-aware Federated learning (PaFL) as a paradigm which uses personalization objectives during FL training and allows them to vary across rounds. Our preliminary results show a $50\%$ reduction in underperforming clients for the language task with knowledge distillation. For the image task, PaFL with elastic weight consolidation or knowledge distillation avoids doubling the number of underperformers. Thus, we argue that PaFL represents a more promising means of reducing the need for personalization.

## 1 INTRODUCTION

Edge devices represent a pool of computational power and data for ML tasks. To use such devices while minimizing communication costs, McMahan et al. (2017) introduced Federated Learning (FL). Federated Learning trains models directly on clients devices without sharing data. As the data distribution differs across clients, FL must balance average performance and performance on specific clients. In some cases, a federated model may perform worse than a fully local one—thus lowering the incentive for FL participation.

The existing body of work on balancing global and local performance focuses on two primary means of improving the client accuracy distribution. Li et al. (2019a) and Li et al. (2020a) propose two Fair FL techniques, q-Fair Federated Learning (q-FFL) and Tilted Empirical Risk Minimization (TERM), which raise the accuracy of the worst-performers by focusing on clients with large losses during global FL training. Alternatively, using *local adaptation* (personalization) methods such as Freezebase (FB), Multi-task Learning (MTL) with Elastic Weight Consolidation (EWC), and Knowledge Distillation (KD) has been recommended by Yu et al. (2020) and Mansour et al. (2020) in order to construct effective local models from the global one. Since personalization is local, the natural baseline of comparison is a local model trained only on the client. In this work, *relative accuracy* refers to the accuracy difference between a federated and local model on a client test set.

While the sets of potential use cases for fairness and personalization are not identical—e.g. personalization would be inappropriate for very low-data clients—FFL could potentially construct a fairer relative accuracy distribution without hurting average performance. For FFL to reduce the need for personalization it would have to lower the number of underperforming clients or improve their av-

erage relative accuracy enough to require *less* adaptation. This is not what we observe in practice, as our experiments show FFL to have either a negative impact on relative accuracy or none at all.

Our contribution is threefold:

1. We construct an initial empirical evaluation of the relative accuracy distribution of models trained with FFL on the Reddit, CIFAR-10, and FEMNIST datasets for next word prediction and image recognition tasks. We then compare the number of underperforming clients for fair models to a FedAvg baseline. During our evaluation, we show that FFL does not significantly reduce the number of underperformers or improve the relative accuracy distribution on Reddit and brings little benefit over a combination of FedAvg and local adaptation. Concerningly, it doubled the number of underperforming clients on FEMNIST.

2. We investigate any potential synergies between FFL and personalization by adapting fair federated models. Results show that the adapted models do not significantly outperform those initially trained with FedAvg in average accuracy or number of underperformers.

3. We instead propose Personalization-aware Federated Learning as a paradigm which uses local adaptation techniques during FL training. Preliminary experimental results on the language task show a significant reduction in the number of underperforming clients over FFL when applying KD after model convergence without any downsides to subsequent local adaptation. PaFL can also avoid the increase in the number of underperforming clients observed for image recognition on FEMNIST when using EWC or KD. We speculate that PaFL outperforms the loss-based weighted averaging mechanism of FFL because it can take advantage of data from atypical clients without greatly harming average performance.

## 2 BACKGROUND AND RELATED WORK

**Statistical (data) heterogeneity** Data generation and accrual naturally vary across devices. Factors such as sensor characteristics, geographic location, time, and user behaviour may influence the precise deviations in data distribution seen by a client—e.g. feature, label, or quantity skew as reported by Kairouz et al. (2019, sec.3.1)—, which in turn prevents treating client data as Independent and Identically Distributed (IID). Non-IID data has been shown to impact both global accuracy (Zhao et al., 2018; Hsieh et al., 2020) and theoretical convergence bounds (Li et al., 2019b).

**System (hardware) heterogeneity** Devices within the federated network may differ in computational ability, network speed, reliability and data-gathering hardware. System heterogeneity creates barriers to achieving a fault and straggler-tolerant algorithm. However, Li et al. (2019b) indicate that it behaves similarly to data heterogeneity during training and benefits from similar solutions.

### 2.1 FAIR FEDERATED LEARNING

The standard objective function of FL is formulated by Li et al. (2020b) as seen in Eq. (1)

$$\min_w f(w) = \sum_{k=1}^{m} p_k F_k(w) \, , \tag{1}$$

where $f$ is the federated loss, $m$ is the client count, $w$ is the model at the start of a round, and $F_k$ is the loss of client $k$ weighted by $p_k$. For a total number of training examples $n$, $p_k$ is defined as the proportion of examples on the client $\frac{n_k}{n}$ or as the inverse of the number of clients $\frac{1}{m}$. The Federated Averaging (FedAvg) algorithm introduced by McMahan et al. (2017) optimizes the objective by training locally on clients and then summing the parameters of each model $G_k$ weighted by $p_k$ into an update to the previous model $G_t$ using an aggregation learning rate $\eta$, as seen in Eq. (2)

$$G^{t+1} = G^t + \eta \left( \sum_{k=1}^{m} p_k G_k^t \right) \, . \tag{2}$$

Li et al. (2019a) propose Fair Federated Learning (FFL), which attempts to train a model with a better accuracy distribution. They define q-FFL as a version of FFL with the objective from Eq. (3)

$$\min_w f(w) = \sum_{k=1}^{m} \frac{p_k}{q+1} F_k^{q+1}(w) \, , \tag{3}$$

where $q$ controls the degree of desired fairness. A value of $q = 0$ corresponds to FedAvg while larger values prioritize clients with higher losses. As $q$ approaches infinity, the objective function approaches optimizing solely for the highest-loss client.

Li et al. (2020a) develop a more general technique shown in Eq. (4) which behaves similarly to q-FFL when applied for FL and can be tuned using $t$. It is important to note that $t$ and $q$ do not scale fairness at the same rate and need to be optimized independently. While the two objectives show a comparable accuracy distribution improvements in the evaluations of Li et al. (2020a), it is unclear how they would affect the *relative accuracy* distribution.

$$\min_w f(w) = \frac{1}{t} \log(\sum_{k=1}^{m} p_k e^{tF_k(w)}) \, , \tag{4}$$

The most relevant recent FFL work is Ditto published by Li et al. (2021) as a means of constructing personalized models while encouraging fairness for the global model. Ditto functions by training a local personalized model in parallel with the global one. It keeps them in sync by making the local model minimize the $L2$ distance to the federated model similarly to the personalization techniques discussed bellow. Li et al. (2021) show it to be superior to TERM, however it is not fit for our purposes as it replicates the costs and limitations of local adaptation on every round.

## 2.2 LOCAL ADAPTATION

The analysis of Yu et al. (2020) established that not only does the federated model perform worse on heterogeneous clients, as previously noted by Li et al. (2019a); Kairouz et al. (2019), but it may offer inferior performance to local models. Thus, certain clients may receive no benefit from participating in the FL process. Yu et al. (2020) recommend addressing this accuracy gap via personalization methods using the federated model as a starting point for local training.

**Multi-task Learning (MTL)**    The task of the global model is to perform well on the distributions of all clients while a local one must perform on the distribution of a single client. Equation (5) frames this as a Multi-task Learning (MTL) problem using the Elastic Weight Consolidation technique (EWC) introduced by Kirkpatrick et al. (2017) to mitigate Catastrophic forgetting (Goodfellow et al., 2013)

$$l(C, x) = L(C, x) + \sum_i \frac{\lambda}{2} M[i](C[i] - G[i])^2 \, , \tag{5}$$

where L is the client loss, $\lambda$ determines the weighting between the two tasks and $M$ is the Fisher information matrix.

**Fine-tuning (FT) and Freezebase (FB)**    When a client receives a global model after the FL process, it can apply Fine-tuning (see Wang et al. (2019); Paulik et al. (2022) and Mansour et al. (2020, Section D.2)) to retrain the model on its data. To avoid potential Catastrophic forgetting, Yu et al. (2020) also opt to apply Freezebase (FB) as a variant of FT which retrains only the top layer.

**Knowledge Distillation (KD)**    As an alternative to EWC and FT, Knowledge Distillation (Hinton et al., 2015) uses the global model as a teacher for a client model. For the pure logit outputs of the federated model $G(x)$ and client model $C(x)$, the client minimizes the loss seen in Eq. (6)

$$l(C, x) = \alpha K^2 L(C, x) + (1 - \alpha) K_L(\sigma(G(x) / K), \sigma(C(x) / K)) \, , \tag{6}$$

where $L$ is the client loss, $K_L$ is the Kullback-Leibler divergence (Kullback & Leibler, 1951), $\sigma$ is the softmax of the logit output, $\alpha$ is the weighting of the client loss and $K$ is the temperature.

## 3 METHODS

### 3.1 PERSONALIZATION-AWARE FEDERATED LEARNING

As an alternative to FFL for reducing personalization costs, we present a method based on modifying the local loss function during FL training. Federated learning and local adaptation have historically

been regarded as largely separated phases. One means of combining them is to allow personalization methods which operate purely by modifying the local client objective function to be used at each round prior to aggregation. This work uses the term *Personalization-aware Federated Learning (PaFL)* to refer to such a paradigm. The FedProx algorithm developed by Li et al. (2018) may be considered prototypical as it injects the $L2$ norm of the model weight differences into the loss function of clients. However, their motivation was to improve convergence rather than local performance and their loss did not take into account the *importance* of each model weight.

PaFL extends the principle behind FedProx by allowing the personalization method and its weighting to vary across rounds. Beyond improved convergence, such a process may bring benefits to the final locally-trained models by providing continuity in the local objective between FL training and the final adaptation stage if the same loss function is used. Additionally, there is reason to believe that PaFL could be more powerful conceptually than q-FFL (Eq. (3)) and TERM (Eq. (4)). Loss-based weighted averaging has no means beyond averaging of reconciling differences between models required by clients with equally high losses and highly dissimilar data partitions. Additionally, q-FFL and TERM do not attempt to take the client's relation to the global distribution—beyond the current round—into account. By contrast, PaFL allows clients for whom the global model performs badly to diverge in a manner which maintains accuracy on the whole federated distribution.

Formally, PaFL can be defined as a type of Federated Learning with a potential additional training round at the end which allows clients to keep their locally trained model—representing the personalization phase in standard FL. Each client has a loss function obeying the following structure:

$$l(C, x, t) = \mu(t) L(C, x) + (1 - \mu(t)) D(t)(C, G^y, x) , \qquad (7)$$

where $t$ is the current round, $L(C, x, t)$ is the training loss and $D(t)$ returns a personalization loss function for the current round—potentially dependent on the data point $x$. The weight of each term is set per round through the weighting function $\mu(t)$.

## 3.2 FFL MODIFICATIONS

Two implementation details of q-FedAvg—the q-FFL implementation proposed by Li et al. (2019a)—are worth discussing. First, the choice of $q$ impacts the optimal aggregation learning rate $\eta$. Rather than determining $\eta$ for each $q$, the authors optimize it for $q = 0$ and then scale it for $q > 0$. Since task training parameters are already optimized for FedAvg (i.e., q-FedAvg with $q = 0$) in all tasks, we do not change $\eta$. Second, the original publication (Li et al., 2019a) uses weighted sampling of devices based on their share of the total training examples, followed by uniform averaging. This methodology is untenable in many real-world scenarios, as it would require a server being aware of the amount of data available in each client a priori. Thus, we use uniform sampling and weighted averaging. The same considerations are applied to the TERM equivalent of q-FedAvg.

## 4 EXPERIMENTAL DESIGN

### 4.1 TASKS

Following the lead of Yu et al. (2020) and McMahan et al. (2017), we train models using FedAvg, q-FedAvg, TERM, or PaFL for next-word prediction on a version of the Reddit (Caldas et al., 2018) dataset with 80 000 participants and for image recognition. For image recognition we use CIFAR-10 partitioned into 100 participants as well as the naturally heterogeneous Federated Extended MNIST (FEMNIST) (Caldas et al., 2018) dataset. For PaFL we choose a simple proof-of-concept training sequence where we apply KD or EWC with constant weightings and parameters after the model has converged at the halfway point of training—denoted $H_{EWC}$ and $H_{KD}$. In order to allow direct comparison against previous work, the training pipelines and model architectures of Yu et al. (2020) and Caldas et al. (2018) were adapted. As such, we only note necessary changes from the initial works and provide the full details necessary for reproducibility in Appendix A.2. For both FFL methods we tune their fairness hyperparameter and report performance for values which exhibit relevant behaviour. On FEMNIST we do not tune the value of $t$ for TERM and instead reuse the $t = 1$ value chosen by Li et al. (2021). In the case of PaFL, we use the same parameters for the losses after the halfway point as in the local adaptation setup (Appendix A.2).

**Next word prediction**   During the FL training process $\approx 5\%$ of the federated test set is used to track convergence, with the full test set being used for the final evaluation after training. Federated models are trained for $1\,000$ rounds using 20 clients per round, rather than the $5\,000$ rounds of 100 clients used by Yu et al. (2020). This will be shown to be sufficient for the baseline FedAvg model to reach an accuracy of $17.8\%$, which is close to the original optimum of $19.29\%$. Only $\sim 65\,500$ clients are evaluated and $\sim 18\,500$ clients adapted due to resource constraints (Appendix A.1).

**FEMNIST image classification**   Rather than subsampling $5\%$ of the data, we keep the first 350 clients with more than 10 data points out of the total $3\,597$ clients in the FEMINIST dataset. Since we require both local and federated testing sets, we use $70\%$ of a client's data for training, $10\%$ for local testing, and add the remaining $20\%$ to the federated test set. For the FL process, we use an aggregation learning rate of $\eta = 1.0$ with 10 clients per round for 500 rounds, instead of the 3 clients per round used by Caldas et al. (2018). For each client, we use SGD with a learning rate of $0.1$ and a batch size of 32. During adaptation, we lower the learning rate to $0.01$. The CIFAR setup remained unchanged from Yu et al. (2020).

## 4.2   PERFORMANCE EVALUATION

**Federated and absolute local performance**   The first set of experiments intends to compare the accuracy on the federated test and the local accuracy distribution of models trained with FFL methods or PaFL. Too large of a drop in federated performance may cause a certain fairness level or PaFL configuration to be considered overall unusable. For a given model to perform well locally, it must not bring noticeable harm to average local accuracy while reducing variance when compared to FedAvg. If local training and adaptation are unfeasible on an underperforming client due to lacking data or computational power, FFL or PaFL could still allow for improvements.

**Relative local performance**   The second set involves assessing the difference in accuracy between federated or adapted models and purely local ones when evaluated on client data. The most important factors for the relative utility of such models are the number of clients which receive an improvement and the average population improvement. For local adaptation, FFL, or PaFL configuration to be worthwhile it must increase the number of clients which benefit while maintaining or improving the average accuracy difference. If a synergistic existed between FFL or PaFL and local adaptation, models trained using such techniques would either receive a larger improvement in average accuracy or result in a lower number of underperforming clients after adaptation.

## 5   RESULTS

### 5.1   FFL BASELINE PERFORMANCE

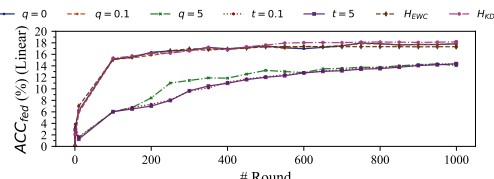 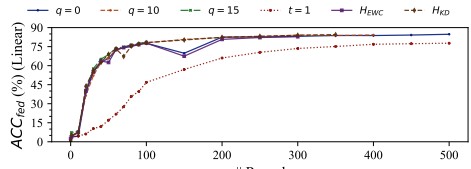

(a) Language task federated performance, TERM always harms performance while q-FFL only does so for $q \geq 1.0$ and only substantially at the shown $q = 5$—included to demonstrate the impact of increasing fairness. Both $H_{EWC}$ and $H_{KD}$ perform close to the FedAvg baseline with $H_{KD}$ exceeding it.

(b) FEMNIST federated performance, q-FFL causes increased instability in the training process and outright divergence past a certain round. Our proposed technique also diverges for both $H_{EWC}$ and $H_{KD}$, however it reaches a similar accuracy to FedAvg ($q = 0$) prior to doing so. TERM performs acceptably as it does not diverge.

Figure 1: Federated accuracy of models across rounds on Reddit and FEMNIST.

To establish a performance baseline, Fig. 1a provides an overview of the convergence process for next-word prediction on Reddit while Table 1 showcases summary data for federated performance

and absolute local performance. From Fig. 1a it can be seen that the impact of q-FFL accuracy is neutral to negative while that of TERM is highly negative for all $t$. The fairest q-FedAvg value ($q = 5$) shows a noticeable dip in accuracy. Fairness seems to reliably reduce the accuracy variance for $q \geq 1$, however the performance cost is too large. We were unable to find a $t$-value leading to an acceptable performance for TERM, as such we excluded it from future Reddit experiments.

| Objective | $ACC_{fed}(\%)$ | $Avg_{loc}(\%)$ | $B_{loc}10\%(\%)$ | $W_{loc}10\%(\%)$ | $(Var_{Avg})$ | $(Var_B)$ | $(Var_W)$ |
|---|---|---|---|---|---|---|---|
| $q = 0$ | 17.826 | 18.645 | 24.572 | 14.815 | 9.177 | **22.114** | 1.072 |
| $q = 0.1$ | 17.789 | 18.66 | 24.843 | 14.728 | 9.81 | 22.914 | 1.036 |
| $q = 5$ | 14.056 | 14.819 | 20.208 | 11.66 | **7.983** | 26.769 | 0.69 |
| $t = 0.1$ | 14.299 | 16.476 | **28.985** | 11.806 | 39.584 | 176.42 | **0.369** |
| $t = 5$ | 14.373 | 16.438 | 28.981 | 11.766 | 39.642 | 175.96 | 0.382 |
| $H_{EWC}$ | 17.322 | 18.255 | 24.653 | 14.226 | 10.277 | 23.059 | 1.185 |
| $H_{KD}$ | **18.177** | **19.179** | 26.406 | **14.887** | 12.438 | 25.85 | 1.039 |
| Local | NaN | 4.456 | 10.227 | 1.204 | 8.777 | 31.11 | 0.893 |

Table 1: Results showing the federated and absolute local performance on Reddit. While fairness does decrease variance at $q \geq 1.0$, the harm to accuracy is too great. The proposed $H_{KD}$ model improves accuracy across clients but increases variance for everyone except the worst performers.

Figure 1b and Table 2 show FEMNIST image recognition models to be highly sensitive to $q$ when trained with q-FedAvg or either PaFL version as their performance oscillates repeatedly or diverges. For such models, we test and adapt its last version prior to divergence. Unlike Reddit, nothing resembling a trend emerges for any metric as fairness increases. Nevertheless, a good balance between performance and average variance is struck by $q = 10$—except for the fairly high worst-performer variance. Unlike the language task, TERM is sufficiently well-behaved on this dataset for models trained using it to be used in later adaptation experiments.

| Objective | $ACC_{fed}(\%)$ | $Avg_{loc}(\%)$ | $B_{loc}10\%(\%)$ | $W_{loc}10\%(\%)$ | $(Var_{Avg})$ | $(Var_B)$ | $(Var_W)$ |
|---|---|---|---|---|---|---|---|
| $q = 0$ | **84.739** | 75.341 | 99.435 | 35.717 | 432.507 | 1.151 | 73.747 |
| $q = 10$ | 84.19 | **76.591** | 99.013 | **42.055** | **320.385** | 1.714 | 64.049 |
| $q = 15$ | 78.634 | 69.749 | 96.681 | 34.988 | 374.637 | 5.177 | **47.761** |
| $t = 1$ | 77.706 | 69.134 | 98.628 | 33.478 | 417.448 | 2.771 | 56.543 |
| $H_{EWC}$ | 82.825 | 73.964 | 99.321 | 33.457 | 465.605 | 1.376 | 71.481 |
| $H_{KD}$ | 84.51 | 75.243 | **99.491** | 34.34 | 443.015 | **1.07** | 62.91 |
| Local | NaN | 46.322 | 92.848 | 0.0 | 1006.77 | 18.144 | 0.0 |

Table 2: Results for FEMNIST. Given the much larger variability in performance compared to Reddit, the only acceptable fair model is the one trained with $q = 10$. Note that the average accuracy of the best-performing local clients is close to that of the best-performing partitions for the federated model. Unlike the language task, using KD during FL seems to primarily help the best performers.

Table 3 indicates the CIFAR-10 image classification task to be more resilient to fairness than previous tasks, with only a small accuracy decrease being noticeable in models trained using $q \geq 10$. The lower sensitivity of this task to training parameters is consistent with the previous findings of Yu et al. (2020) on differentialy private FL and robust FL. Due to the similarity in results across fairness levels, the convergence graph for CIFAR-10 was relegated to the appendix (Fig. 3). Given the lack of insight, we chose not to expand the CIFAR-10 experiments past q-FedAvg and $H_{KD}$. Overall these findings indicate that the dataset heterogeneity must be meaningful rather than artificially imposed for significant effects to emerge for either FFL or PaFL.

## 5.2 FFL FAILS TO IMPROVE RELATIVE PERFORMANCE

Having established baselines of accuracy for fair models, we can now evaluate the local relative performance of FFL, PaFL and their interactions with local adaptation. The CIFAR-10 data (Table 7) is unsuitable for multiple reasons including its artificial partitioning and the fact that the federated model does not benefit from local adaptation as it outperforms a local one for all clients.

| Objective | $ACC_{fed}(\%)$ | $Avg_{loc}(\%)$ | $B_{loc}10\%(\%)$ | $W_{loc}10\%(\%)$ | $(Var_{Avg})$ | $(Var_B)$ | $(Var_W)$ |
|---|---|---|---|---|---|---|---|
| $q = 0$ | 81.28 | **81.37** | **82.255** | **79.864** | 0.568 | 0.022 | 1.067 |
| $q = 5$ | **81.86** | 81.221 | 82.011 | 79.794 | **0.446** | **0.004** | **0.643** |
| $q = 15$ | 78.16 | 79.935 | 81.178 | 77.885 | 0.945 | 0.02 | 1.267 |
| Local | NaN | 31.718 | 38.297 | 24.649 | 16.3 | 1.543 | 0.906 |

Table 3: Results for CIFAR-10. Unlike the language task (Table 1), $q = 5$ represents a clear optimum in terms of variance while maintaining performance. However, differences between all models are very small and cannot be guaranteed to be significant (see Appendix A.2).

For q-FFL, the results for the language task showcased in Table 4 are less than satisfactory as fair models fail to provide benefits in terms of the number of underperforming clients, relative accuracy, or variance on average. Furthermore, fair models do not offer improved accuracy once adapted—this is directly visible in the Fig. 2a scatter plot of relative accuracy against fully local accuracy.

| Objective | Adapt | $Avg(\%)$ | $\% <0$ | $B10\%(\%)$ | $W10\%(\%)$ | $(Var_{Avg})$ | $(Var_B)$ | $(Var_W)$ |
|---|---|---|---|---|---|---|---|---|
| $q = 0$ | $q = 0$ | 14.185 | 53 | 20.715 | 9.392 | **13.323** | 34.379 | 20.201 |
| | A_FB | 15.87 | 0 | 25.849 | **11.311** | 29.216 | 149.736 | 3.246 |
| | A_MTL | **16.046** | 0 | 27.558 | 11.304 | 36.067 | 178.387 | 3.337 |
| | A_KD | 15.538 | 0 | 24.376 | 11.209 | 23.112 | 115.016 | 3.183 |
| $q = 0.1$ | $q = 0.1$ | 14.208 | 50 | 20.907 | 9.359 | 13.742 | 35.005 | 20.733 |
| | A_FB | 15.827 | 0 | 25.964 | 11.261 | 29.505 | 149.011 | 3.212 |
| | A_MTL | 15.839 | 0 | 27.692 | 11.024 | 37.108 | 179.066 | 3.336 |
| | A_KD | 15.546 | 0 | 24.614 | 11.19 | 23.95 | 118.471 | 3.166 |
| $H_{EWC}$ | $H_{EWC}$ | 13.807 | 108 | 20.723 | 8.681 | 14.994 | 38.119 | 24.938 |
| | A_FB | 15.423 | 0 | 25.709 | 10.88 | 29.795 | 149.308 | 3.02 |
| | A_MTL | 15.561 | 0 | 27.482 | 10.762 | 37.251 | 179.528 | 3.266 |
| | A_KD | 15.157 | 0 | 24.336 | 10.823 | 23.996 | 117.427 | 3.041 |
| $H_{KD}$ | $H_{KD}$ | 14.729 | 27 | 22.038 | 9.971 | 14.969 | 41.225 | 18.409 |
| | A_FB | 15.772 | 0 | 26.533 | 11.154 | 31.068 | 150.552 | **3.156** |
| | A_MTL | 15.824 | 2 | **28.217** | 10.966 | 38.439 | 178.543 | 3.469 |
| | A_KD | 15.698 | 0 | 25.358 | 11.214 | 25.367 | 119.418 | 3.241 |

Table 4: Relative performance on the Reddit dataset of the acceptable fair models and our PaFL variants. The best value in a column is in bold while the best in a group is underlined. The chosen optimal fair model does not significantly reduce the number of underperformers. Alternatively, $H_{KD}$ lowers it to half. Local adaptation provides simillar results regardless of starting point.

The results for image recognition on FEMNIST are more unusual yet similarly discouraging for both q-FFL and TERM. Table 5 makes it clear that the fair model actually achieves a higher relative accuracy on average and amongst the top $10\%$ of performers at the cost of obtaining a *negative* average accuracy on the relative worst performers. Additionally, it has more than twice as many clients with negative relative accuracies. We speculate that since the federated model has a lower absolute accuracy variance, it cannot obtain a good enough level of performance on clients which are able to train high-quality local models. This is corroborated by the final distribution shown in Fig. 2b, as nearly *all* the underperforming clients have high local model accuracy. Another factor to consider is the atypical behaviour of personalization on FEMNIST and CIFAR-10. Models trained with FedAvg and then adapted tend to converge to nearly the same relative accuracy regardless of the specific adaptation technique. Thus, FedAvg is potentially near-optimal for the dataset model.

## 5.3 PaFL as an Alternative

Having shown the inability of FFL to replace or enhance local adaptation, we argue that it is not the right approach for this domain. In principle, for an FL algorithm to provide benefits in terms of relative accuracy it must achieve two goals. First, it must make sure that the worst-performing clients receive a sufficient level of accuracy to match or exceed local models. Second, for the clients with the *best* local models, it must provide disproportionately high accuracy. While FFL may help

| Objective | Adapt | $Avg(\%)$ | $\% <0$ | $B10\%(\%)$ | $W10\%(\%)$ | $(Var_{Avg})$ | $(Var_B)$ | $(Var_W)$ |
|---|---|---|---|---|---|---|---|---|
| $q=0$ | $q=0$ | 29.02 | 16 | 65.768 | 2.729 | 338.387 | 137.366 | 12.33 |
| | A_FB | 28.954 | 17 | 65.463 | 2.72 | 334.754 | 134.824 | 11.853 |
| | A_MTL | 28.994 | 16 | 65.672 | **2.802** | 336.25 | 137.507 | 11.325 |
| | A_KD | 28.986 | 16 | 65.684 | 2.788 | 337.14 | 137.796 | 11.594 |
| $q=10$ | $q=10$ | **30.269** | 39 | **79.687** | -14.844 | 673.351 | **111.144** | 206.995 |
| | A_FB | 28.613 | **12** | 64.818 | 2.699 | 320.729 | 123.679 | 15.552 |
| | A_MTL | 28.612 | 14 | 64.818 | 2.516 | 321.3 | 123.679 | 15.869 |
| | A_KD | 28.563 | 14 | 64.645 | 2.52 | 320.957 | 127.618 | 15.934 |
| $t=1$ | $t=1$ | 22.812 | 56 | 73.593 | -17.069 | 627.167 | 215.261 | 91.294 |
| | A_FB | 21.261 | 15 | 50.057 | 0.806 | **201.35** | 127.802 | 8.417 |
| | A_MTL | 21.362 | 15 | 50.218 | 0.765 | 201.703 | 123.192 | 8.102 |
| | A_KD | 21.202 | 15 | 50.1 | 0.706 | 202.488 | 125.153 | **7.71** |
| $H_{EWC}$ | $H_{EWC}$ | 27.642 | 13 | 62.157 | 1.522 | 315.388 | 123.438 | 19.225 |
| | A_FB | 27.558 | 14 | 62.431 | 1.404 | 316.622 | 124.092 | 18.888 |
| | A_MTL | 27.603 | 13 | 62.588 | 1.349 | 315.619 | 122.717 | 20.502 |
| | A_KD | 27.611 | 13 | 62.588 | 1.542 | 314.112 | 122.717 | 19.176 |
| $H_{KD}$ | $H_{KD}$ | 28.921 | 16 | 66.178 | 1.65 | 353.698 | 154.217 | 16.231 |
| | A_FB | 28.916 | 17 | 66.605 | 1.644 | 350.631 | 152.111 | 16.922 |
| | A_MTL | 28.871 | 16 | 66.605 | 1.719 | 350.742 | 152.111 | 16.865 |
| | A_KD | 28.967 | 15 | 66.329 | 1.869 | 351.991 | 150.583 | 17.124 |

Table 5: FEMNIST performance of the best fair model and our proposed alternative to FFL. Despite providing the highest average accuracy and accuracy amongst the best $10\%$, the model trained using $q = 10$ has more than double the number of underperformers of FedAvg ($q = 0$)—as does TERM with $t = 1$. For PaFL, $H_{KD}$ is close to FedAvg while $H_{EWC}$ brings a trivial improvement.

(a) Language task results, fairness shows no benefit while $H_{KD}$ reduces the number of underperformers.

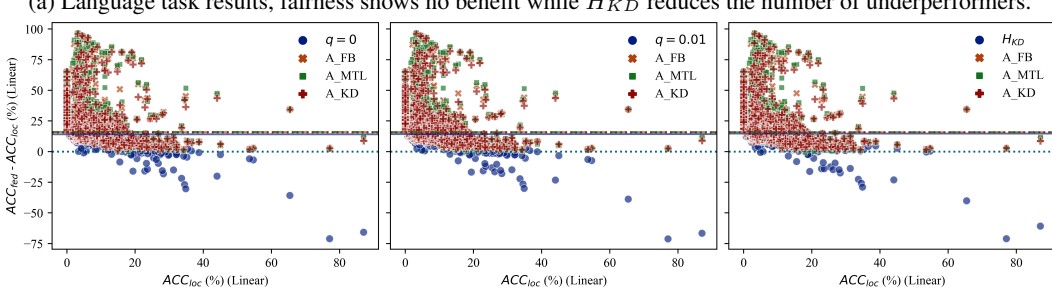

(b) FEMNIST results, clients with highly accurate local models are underserved by federated models trained using $q = 10$. Alternatively, those trained using FedAvg or $H_{EWC}$ achieve nearly identical performance.

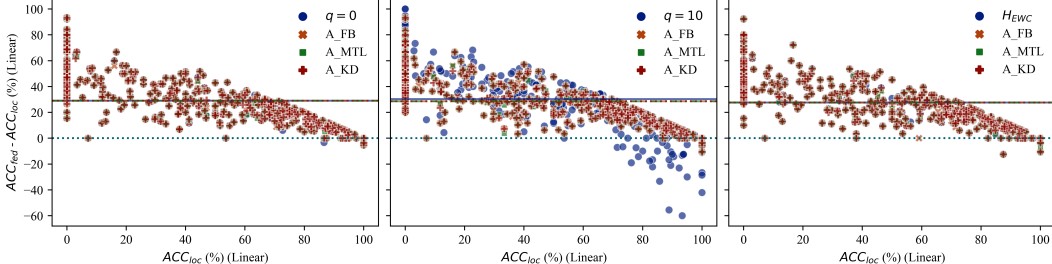

Figure 2: Federated model relative accuracy on a client plotted against local client model accuracy.

fulfil the first requirement, its inability to raise the floor of the worst performers without hurting the ceiling of those that might have a good local model makes it incapable of fulfilling the second.

PaFL in the most general case offers an alternative where models can be kept closer to one another during training and only allowed to diverge in ways which hurt federated performance the least. Unlike blind regularization based on the norm of the distance between model parameters (e.g. Fed-Prox), EWC and KD offer the distinct advantage of determining how a parameter may diverge based on its importance to federated performance. Thus, the model can learn from highly heterogeneous data and raise its accuracy floor for the worst-performers without hurting the accuracy ceiling of the best or even improving it. The models trained with either EWC or KD past the halfway point, $H_{EWC}$ and $H_{KD}$, have already been included in previous tables and graphs to allow for comparison.

Preliminary results for the language task are promising in the case of $H_{KD}$, Fig. 1a and Table 1 indicate that it performs better than FedAvg and FFL models in every metric except average and best-performer variance. Importantly, variance is not increased for the worst performers. While $H_{EWC}$ is not far below the FedAvg baseline, it fails to provide any obvious improvements. In terms of relative accuracy, Table 4 shows that $H_{KD}$ halves the number of underperformers and provides the best average relative accuracy. However, this higher baseline does not translate to improved relative accuracy for adapted models. Overall, by lowering the number of clients which *require* adaptation in order to receive an incentive to participate $H_{KD}$ successfully reduces the need for personalization on Reddit. On the other hand, $H_{EWC}$ seems to double the number of underperforming clients for the fixed chosen $\lambda$ although a different value or scheduling across rounds may change results.

For image recognition on FEMNIST, $H_{KD}$ and $H_{EWC}$ are satisfactory in terms of federated and average accuracy according to Fig. 1b and Table 2. On the other hand, the results related to relative accuracy shown in Table 5 are mixed. While they both avoid the doubling in underperforming clients that fair models suffer, locally adapted models starting from $H_{KD}$ as a baseline do not seem to outperform those adapted from FedAvg. Perhaps surprisingly given its failure on the language task, $H_{EWC}$ brings a very small reduction to the number of underperformers for baseline and adapted models. While this is not sufficient to draw strong conclusions, it does indicate that PaFL configurations behave differently across domains.

Overall, for both tasks PaFL variants have brought small to medium improvements to the number of underperforming clients, average relative accuracy and associated metrics without clear downsides. Nonetheless, more experimentation is required on the specific parameters and on other techniques originating from Multi-task Learning and its associated fields.

# 6 CONCLUSION

This paper set out to find a means of reducing the amount of personalization needed to incentivize FL participation for clients whose local model outperforms a federated one. Such a reduction would be relevant for federated networks containing devices with limited capabilities for retraining or little data. Our investigation began with Fair Federated Learning due to the possibility that reducing disparity in the local accuracy distribution would translate to reducing disparity in the relative accuracy distribution. Our experimental results indicate that FFL is unlikely to provide the desired properties as it has been shown to maintain the number of underperformers on our language task while increasing it by more than $100\%$ for image recognition on FEMNIST. We speculate that the reason for this is that although it can help clients for which the global model is highly inaccurate, it cannot help those for whom relative underperformance is caused by an extremely accurate local model. As an alternative, Personalization-aware Federated Learning allows loss functions historically used for local adaptation to be applied during FL and to vary across rounds. We tested applying EWC or KD after the model had partially converged in the hopes that it would allow it to learn from worst-performer data without sacrificing performance on the federated distribution or best performers. While our chosen EWC configuration did not bring a meaningful improvement over FedAvg on the language task, KD showed promising results by reducing the number of underperformers by up to $50\%$. Both of them avoided increasing the number of underpeformers on the FEMNIST image recognition task. Unlike more complex systems which simultaneously train local and federated models during FL, this approach has little computational overhead. Consequently, we recommend using PaFL to incentivize FL participation without explicit local adaptation and advise further research adapting research directions from the field of Multi-task learning to FL.

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

# A APPENDIX

## A.1 HARDWARE

Each node of the cluster that the experiments were run on holds four Nvidia A100 GPUs with 80GiB VRAM and 2 x AMD EPYC 7763 64-Core processors alongside a 1000GiB shared memory. Given the quotas and service levels of the cluster, the number of clients on which the federated model could be tested locally for the language task was limited to $\sim 65\,500$ while the number that could be adapted was limited to ($\sim 18500$)—the entire client set was available during FL training. To compensate for this fact, all charts and tables comparing local model performance or adaptation performance only use data from the client-set common to all results. This limitation did not impact any other sets of experiments.

## A.2 FULL EXPERIMENTAL DETAILS

During local adaptation we reuse the parameters recommended by Yu et al. (2020). MTL uses a weighting of $\lambda = 5000$ while KD uses a temperature $k = 6$ and weighting $\lambda = 0.95$,

**Next-word prediction** A standard LSTM with 2 layers, 200 hidden units and 10 million parameters is used to predict the next word in a sentence for each client used during training or testing. We reuse the dictionary of the $50\,000$ most frequent words compilled by Yu et al. (2020), all other words are replaced with a placeholder. The first $90\%$ of a users posts, chronologically, are used as a training set with the final $10\%$ being reserved for local testing. A separate federated testing set is maintained for evaluating global task performance, during the FL training process $\approx 5\%$ of it is used to track convergence with the full test-set being used for the final evaluation after training. Federated models are trained for $1\,000$ rounds using 20 clients per round rather than the $5\,000$ rounds of $1\,000$ clients used by Yu et al. (2020). On the client side, each model is trained for 2 internal epochs with a batch size of 20 using Stochastic Gradient Descent with the learning rate set to 40. For adaptation, we use a learning rate of 1 and batch size of 20 for 100 epochs of retraining. Only $\sim 18\,500$ clients are adapted due to resource constraints (Appendix A.1).

**CIFAR-10 image classification** Since CIFAR-10 is not a naturally federated dataset, a Dirichlet distribution ($\alpha = 0.9$) is used to simulate a non-iid partitioning (Hsu et al., 2019; Yu et al., 2020). A ResNet-18 model is trained over $1,000$ rounds with 10 clients per round. Clients are trained using a batch size of 32 with 2 internal epochs and a learning rate of $0.1$. Test accuracy is computed by multiplying a client's per-class accuracy on the CIFAR-10 test set with its proportion of the local device data. For adaptation, we use a learning rate of $10^{-3}$ and batch size of 32 for 200 epochs of retraining. Training uses SGS with momentum $0.9$ and weight decay $0.0005$,

**FEMNIST image classification** We use a similar experimental setup to Caldas et al. (2018) with a simple two-layer CNN while changing the way the dataset is divided and the FL training parameters. Rather than subsampling $5\%$ of the data, we keep the first 350 with more than 10 data points out of the total $3\,597$ clients in the FEMINIST dataset. Since we require both local and federated testing sets we keep $70\%$ of a clients' data for training, $10\%$ for local testing and add the remaining $20\%$ to the federated test set. For the FL process, we use an aggregation learning rate of $\eta = 1.0$ with 10 clients per round for $1\,000$ rounds instead of the 3 clients per round used by Caldas et al. (2018). For each client, we use SGD with a learning rate of $0.1$ and a batch size of 32.

## A.3 FULL Q-FEDAVG RESULTS

| Objective | Adapt | $Avg(\%)$ | $Acc < 0$ | $B10\%(\%)$ | $W10\%(\%)$ | $(Var_{Avg})$ | $(Var_B)$ | $(Var_W)$ |
|---|---|---|---|---|---|---|---|---|
| $q=0$ | $q=0$ | 14.185 | 53 | 20.715 | 9.392 | 13.323 | 34.379 | 20.201 |
| | A_FB | 15.87 | 0 | 25.849 | 11.311 | 29.216 | 149.736 | 3.246 |
| | A_MTL | 16.046 | 0 | 27.558 | 11.304 | 36.067 | 178.387 | 3.337 |
| | A_KD | 15.538 | 0 | 24.376 | 11.209 | 23.112 | 115.016 | 3.183 |
| $q=0.01$ | $q=0.01$ | 13.989 | 106 | 20.458 | 8.793 | 14.421 | 33.874 | 28.638 |
| | A_FB | 15.799 | 0 | 25.7 | 11.238 | 29.093 | 149.929 | 3.228 |
| | A_MTL | 15.833 | 1 | 27.419 | 11.057 | 36.547 | 180.574 | 3.326 |
| | A_KD | 15.495 | 0 | 24.308 | 11.175 | 23.343 | 117.768 | 3.173 |
| $q=0.1$ | $q=0.1$ | 14.208 | 50 | 20.907 | 9.359 | 13.742 | 35.005 | 20.733 |
| | A_FB | 15.827 | 0 | 25.964 | 11.261 | 29.505 | 149.011 | 3.212 |
| | A_MTL | 15.839 | 0 | 27.692 | 11.024 | 37.108 | 179.066 | 3.336 |
| | A_KD | 15.546 | 0 | 24.614 | 11.19 | 23.95 | 118.471 | 3.166 |
| $q=0.5$ | $q=0.5$ | 14.097 | 53 | 20.705 | 9.287 | 13.514 | 34.546 | 20.655 |
| | A_FB | 15.837 | 0 | 25.838 | 11.285 | 29.2 | 148.925 | 3.232 |
| | A_MTL | 15.892 | 0 | 27.555 | 11.133 | 36.531 | 179.11 | 3.26 |
| | A_KD | 15.52 | 0 | 24.42 | 11.186 | 23.402 | 116.556 | 3.199 |
| $q=1$ | $q=1$ | 11.397 | 70 | 17.618 | 7.047 | 12.546 | 37.951 | 19.031 |
| | A_FB | 13.463 | 0 | 22.632 | 9.256 | 25.775 | 136.853 | 1.987 |
| | A_MTL | 13.364 | 4 | 24.519 | 8.993 | 35.506 | 187.701 | 2.376 |
| | A_KD | 13.238 | 0 | 21.567 | 9.245 | 20.821 | 105.756 | 2.016 |
| $q=5$ | $q=5$ | 10.384 | 121 | 16.583 | 5.904 | 12.952 | 39.358 | 21.502 |
| | A_FB | 12.771 | 0 | 21.703 | 8.678 | 24.874 | 133.58 | 1.694 |
| | A_MTL | 12.838 | 6 | 23.37 | 8.548 | 33.262 | 181.204 | 2.146 |
| | A_KD | 12.492 | 0 | 20.608 | 8.614 | 19.704 | 99.431 | 1.695 |

Table 6: Reddit full q-FedAvg results.

| Objective | Adapt | $Avg(\%)$ | $Acc < 0$ | $B10\%(\%)$ | $W10\%(\%)$ | $(Var_{Avg})$ | $(Var_B)$ | $(Var_W)$ |
|---|---|---|---|---|---|---|---|---|
| $q=0$ | $q=0$ | 49.652 | 0 | 56.54 | 43.045 | 15.557 | 0.79 | 2.563 |
| | A_FB | 49.669 | 0 | 56.522 | 43.145 | 15.364 | 0.839 | 2.165 |
| | A_MTL | 49.647 | 0 | 56.489 | 43.064 | 15.468 | 0.829 | 2.267 |
| | A_KD | 49.647 | 0 | 56.527 | 43.066 | 15.545 | 0.789 | 2.076 |
| $q=0.1$ | $q=0.1$ | 49.254 | 0 | 56.078 | 42.839 | 15.408 | 0.858 | 1.94 |
| | A_FB | 49.23 | 0 | 55.997 | 42.758 | 15.492 | 0.987 | 1.923 |
| | A_MTL | 49.257 | 0 | 55.925 | 42.933 | 15.569 | 1.129 | 2.077 |
| | A_KD | 49.248 | 0 | 56.047 | 42.82 | 15.486 | 0.996 | 1.973 |
| $q=1$ | $q=1$ | 49.442 | 0 | 56.297 | 42.865 | 15.614 | 1.053 | 1.991 |
| | A_FB | 49.448 | 0 | 56.308 | 42.812 | 15.729 | 0.98 | 2.423 |
| | A_MTL | 49.425 | 0 | 56.352 | 42.786 | 15.728 | 0.966 | 2.287 |
| | A_KD | 49.439 | 0 | 56.314 | 42.78 | 15.742 | 0.981 | 2.307 |
| $q=5$ | $q=5$ | 49.503 | 0 | 56.25 | 42.864 | 15.583 | 0.755 | 2.07 |
| | A_FB | 49.517 | 0 | 56.315 | 42.974 | 15.617 | 0.77 | 2.091 |
| | A_MTL | 49.522 | 0 | 56.404 | 42.909 | 15.651 | 0.603 | 1.874 |
| | A_KD | 49.548 | 0 | 56.416 | 42.991 | 15.531 | 0.678 | 1.902 |
| $q=10$ | $q=10$ | 48.286 | 0 | 54.949 | 42.033 | 14.417 | 0.608 | 1.956 |
| | A_FB | 48.3 | 0 | 54.948 | 42.096 | 14.275 | 0.424 | 1.847 |
| | A_MTL | 48.306 | 0 | 55.025 | 42.087 | 14.468 | 0.565 | 1.71 |
| | A_KD | 48.297 | 0 | 54.987 | 42.021 | 14.504 | 0.507 | 1.873 |
| $q=15$ | $q=15$ | 48.217 | 0 | 54.745 | 41.842 | 14.449 | 0.585 | 2.434 |
| | A_FB | 48.22 | 0 | 54.718 | 41.883 | 14.482 | 0.619 | 1.859 |
| | A_MTL | 48.202 | 0 | 54.761 | 41.824 | 14.444 | 0.614 | 2.191 |
| | A_KD | 48.193 | 0 | 54.779 | 41.713 | 14.763 | 0.564 | 2.473 |

Table 7: CIFAR-10 full q-FedAvg results.

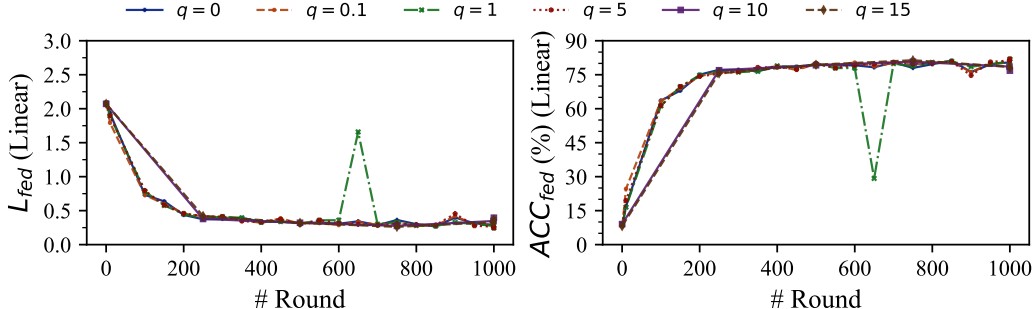

Figure 3: Federated performance of fair models on CIFAR-10. q-FedAvg performs marginally worse for $q \geq 10.0$, however, it must be concluded that the accuracy of the given task is not sensitive enough to fairness to draw strong conclusions.

| Objective | Adapt | $Avg(\%)$ | $Acc < 0$ | $B10\%(\%)$ | $W10\%(\%)$ | $(Var_{Avg})$ | $(Var_B)$ | $(Var_W)$ |
|-----------|-------|-----------|-----------|-------------|-------------|---------------|-----------|-----------|
| $q = 0$ | $q = 0$ | 29.02 | 16 | 65.768 | 2.729 | 338.387 | 137.366 | 12.33 |
| | A_FB | 28.954 | 17 | 65.463 | 2.72 | 334.754 | 134.824 | 11.853 |
| | A_MTL | 28.994 | 16 | 65.672 | 2.802 | 336.25 | 137.507 | 11.325 |
| | A_KD | 28.986 | 16 | 65.684 | 2.788 | 337.14 | 137.796 | 11.594 |
| $q = 0.1$ | $q = 0.1$ | 27.427 | 45 | 80.262 | -23.681 | 761.163 | 117.165 | 165.003 |
| | A_FB | 26.174 | 14 | 59.483 | 2.09 | 282.14 | 128.526 | 14.212 |
| | A_MTL | 26.205 | 14 | 59.567 | 2.345 | 281.689 | 128.659 | 15.377 |
| | A_KD | 26.18 | 15 | 59.613 | 1.997 | 282.946 | 127.796 | 14.284 |
| $q = 1$ | $q = 1$ | 27.011 | 48 | 78.722 | -18.92 | 699.58 | 131.927 | 150.33 |
| | A_FB | 24.999 | 19 | 57.014 | 0.513 | 270.651 | 140.273 | 24.641 |
| | A_MTL | 25.022 | 17 | 57.179 | 0.76 | 271.681 | 136.18 | 23.188 |
| | A_KD | 25.053 | 17 | 57.165 | 0.859 | 270.926 | 135.297 | 22.658 |
| $q = 5$ | $q = 5$ | 25.048 | 46 | 73.939 | -23.702 | 683.322 | 129.091 | 177.272 |
| | A_FB | 25.385 | 18 | 56.625 | 1.758 | 264.374 | 97.084 | 8.057 |
| | A_MTL | 25.468 | 16 | 56.64 | 2.031 | 261.138 | 96.776 | 8.409 |
| | A_KD | 25.437 | 17 | 57.116 | 1.695 | 267.606 | 91.34 | 10.717 |
| $q = 10$ | $q = 10$ | 30.269 | 39 | 79.687 | -14.844 | 673.351 | 111.144 | 206.995 |
| | A_FB | 28.613 | 12 | 64.818 | 2.699 | 320.729 | 123.679 | 15.552 |
| | A_MTL | 28.612 | 14 | 64.818 | 2.516 | 321.3 | 123.679 | 15.869 |
| | A_KD | 28.563 | 14 | 64.645 | 2.52 | 320.957 | 127.618 | 15.934 |
| $q = 15$ | $q = 15$ | 23.427 | 56 | 74.969 | -23.34 | 699.404 | 145.773 | 170.676 |
| | A_FB | 22.008 | 17 | 51.138 | 1.039 | 214.214 | 115.322 | 7.717 |
| | A_MTL | 21.848 | 21 | 51.167 | 0.149 | 222.45 | 115.582 | 9.945 |
| | A_KD | 21.935 | 17 | 51.467 | 0.84 | 216.567 | 112.101 | 8.538 |

Table 8: FEMNIST full q-FedAvg results.

