# OpenReview forum: "Can Fair Federated Learning reduce the need for personalization?"
_ICLR.cc/2023/Conference — Submitted to ICLR 2023_

### Official Review · Reviewer_Wm3o · 2022-10-18

**Confidence:** 3
**Correctness:** 2
**Technical Novelty And Significance:** 2
**Empirical Novelty And Significance:** 2
**Recommendation:** 3

**Clarity, Quality, Novelty And Reproducibility:**

The quality of this work is below the bar. The originality is very limited. The reproducibility is not so satisfactory.

**Strength And Weaknesses:**

Strengths:
- The problem studied in this paper is interesting.
- The analysis in this paper can provide some reference for future fair federated learning studies.

Weaknesses:
- The scope of this paper is quite narrow and the findings are not significant. The study is limited to Q-FFL and only one baseline method is used. Thus, the conclusion drawn from the results may not be universally valid. In addition, the advantage of knowledge distillation over other types of techniques is not verified.
- Using knowledge distillation to teach local models is a quite common strategy (e.g., [1]), thereby the technical contribution of this paper is very limited.
- The coverage of related work is quite limited. The authors should present a more comprehensive review of fair federated learning and personalized federated learning.
- The experimental setting is unclear. It would be a hard task to reproduce the results without sufficient details.

[1] Yu, T., Bagdasaryan, E., & Shmatikov, V. (2020). Salvaging federated learning by local adaptation. arXiv preprint arXiv:2002.04758.

**Summary Of The Paper:**

This paper studies the performance fairness problem of federated learning. The authors empirically show that Q-Fair Federated Learning fails to improve performance fairness. They also show that knowledge distillation is a better way for fair federated learning without additional personalization mechanisms.

**Summary Of The Review:**

This paper has limited contribution, flawed evaluation, and insufficient literature review. Thus, my recommendation is rejection.

---

> ### Author Response · Authors · 2022-11-19
> **Author Responses to Reviewer Wm3o**
>
> Thank you very much for your review, your suggestions were constructive and we believe we have addressed most of them.
>
> 1. We have expanded from Q-FFL also to include a comparison against the newer TERM[1] algorithm, which can fulfil a similar purpose in improving fairness. We also address Ditto [2], a system which simultaneously trains local and federated models on each client for every round in order to provide both fairness and personalization. We explain that its computational requirements make it unsuitable for our goal as they replicate those of local adaptation. In addition, we now show a preliminary comparison between KD and EWC used during federated training as a justification for choosing KD for the language task—although we must mention that additional tuning and verification are necessary.
> 2. The referenced paper uses KD to personalise after federated training rather than using it within the local loss function during FL. Furthermore, we are not aware of any other works that have looked at the impact that applying such a technique between FL aggregation rounds has on the number of clients for whom the global model underperforms relatively to a purely local one.
> 3. We have included a broader overview of FFL and FL personalization, and we have justified our choice of using q-FFL/TERM in conjunction with local adaptation over other potential approaches, such as the one taken by Ditto [2].
> 4. The full experimental setup has been added to the appendix. The main text specifies which previous works the experiments draw upon and the necessary changes to obtain our experimental setup from them.
>
> We hope you find our response to your comments satisfactory and would like to know what you think about the updated work.
>
> [1] Li, T., Beirami, A., Sanjabi, M., & Smith, V. (2020). Tilted empirical risk minimization. arXiv preprint arXiv:2007.01162.
>
> [2]  Li, T., Hu, S., Beirami, A., & Smith, V. (2021, July). Ditto: Fair and robust federated learning through personalization. In International Conference on Machine Learning (pp. 6357-6368). PMLR.

---

### Official Review · Reviewer_AD53 · 2022-10-20

**Confidence:** 3
**Clarity, Quality, Novelty And Reproducibility:** Limited clarity, Limited quality, Lim…
**Correctness:** 3
**Technical Novelty And Significance:** 2
**Empirical Novelty And Significance:** 2
**Recommendation:** 3

**Strength And Weaknesses:**

Strength:
1. There is a good effort to evaluate Q-Fair Federated Learning (Q-FFL) with comprehensive and sufficient analysis. And all of the formulas in the paper are derived from others' papers, rather than serving the methods proposed in the paper.

Weakness:
1. Use too many words to describe relevant knowledge, but not enough to the author's ideas.
2. The whole ‘Method’ part seems to be the details of the experiment and the analysis of the results. I think they should be put in the experimental position rather than the method.
3. The baseline performance occupies too much space and compresses the space for other experiments.
4. Suggest the author apply the knowledge distillation to other federated learning methods to illustrate the universality.
5. Inadequate citation of papers.


**Summary Of The Paper:**

This paper evaluates Q-Fair Federated Learning (Q-FFL) in personalized federated learning with its underperforming clients and proposes using knowledge distillation during FL training. And experiments with different datasets show a 50% reduction in underperforming clients in the language task with no increase for the image task.

**Summary Of The Review:**

Although it is a good effort to evaluate Q-Fair Federated Learning (Q-FFL) with comprehensive and sufficient analysis. But this paper needs to use more words to describe the author’s ideas, not the other’s.

---

> ### Author Response · Authors · 2022-11-19
> **Author Responses to Reviewer AD53**
>
> Thank you very much for your comments. Our work started out as an experimental study seeking to evaluate the interaction between FFL and personalization, thus we focused on presenting the experimental results in as much detail as possible while highlighting the characteristics of the analysed methods. Following your review, we have addressed the mentioned weaknesses:
>
> 1. We have constrained the overall length of the introduction and background to less than three pages. We have also expanded upon our reasoning for why fairness could reduce the need for personalization while providing more detail on our alternative approach. We have renamed our technique to Personalization-aware Federated Learning to emphasise its versatility. In fact, our method allows any loss function suitable for personalization to be injected into the local objective of a client during FL. We also allow for this function and/or its weighting to vary across rounds. The intent behind it is to learn from clients with highly heterogeneous data without harming the accuracy of the rest of the federated network.
>
> 2. We have restructured these sections according to your recommendation by splitting the methods and experimental design.
>
> 3. The size of the baseline performance evaluation has been reduced significantly in order to allow expansion in later sections.
>
> 4. We shall take this into consideration for future work, for now, we have also added examples of using EWC instead of KD for our alternative method.
>
> 5. Could you please elaborate on which works are inadequately cited?
>
> We look forward to hearing your opinion on this adjusted version.

---

### Official Review · Reviewer_j5Ef · 2022-10-25

**Confidence:** 4
**Clarity, Quality, Novelty And Reproducibility:** The paper is not clearly written, and…
**Correctness:** 2
**Technical Novelty And Significance:** 2
**Empirical Novelty And Significance:** 2
**Recommendation:** 3

**Strength And Weaknesses:**

*Strength*
- The idea is interesting and empirical resutls are provided.

*Weakness*

- The paper is not well organized and difficult to follow.

- The contributions are not clear and properly placed in comparison to related literature.

- What is G in equation (2) ?

- What is he proposed/new part in the paper is not clearly mentioned.

- There is no theoretical justification.

- It looks like the main contribution is in Sec 4.3 but that is also not clearly discussed. The paper is not publishable in its current form.

**Summary Of The Paper:**

The authors talk about the idea of utilizing fairness in reducing the need for personalization. The authors have shown experimentally that the fairness doesn't always help in providing a better starting point for personalization. The authors have proposed a knowledge distillation during FL training to improve the local accuracy. Experimental evidences are provided.

**Summary Of The Review:**

Overall, the idea is interesting but it is not executed well. The paper requires major revision before publications. The main idea is to show that fairness it not suitable for personalization and authors have proposed to change the loss function with knowledge distillation.

---

> ### Author Response · Authors · 2022-11-19
> **Author Responses to Reviewer  j5Ef**
>
> Thank you for the comments, we do agree with the need for revisions. Please find our response to your mentioned weaknesses below:
>
> 1. The paper has been restructured with a clearer separation of chapters.
> 2. The contributions are now more heavily emphasised throughout the entire paper, and more context is provided regarding other FFL algorithms or works at the intersection between FFL and personalization, such as Ditto [1].
> 3. The text surrounding the equation now clearly identifies G as either the federated model at round t or t+1 or the model of client k at round t.
> 4. The work is both an experimental study of the interaction between fairness and personalization in FL and a proposal for an alternative solution to the initial problem of reducing the need for personalization. The impact that either FFL or techniques similar to our alternative solution have on the number of clients for whom the federated model underperforms relative to a local one has not been previously assessed to our knowledge. We have expanded our explanation for the alternative method, which we now call Personalization-aware Federated Learning.
> 5. Greater justification is provided for the potential usefulness of fairness as well as the usage of loss functions such as KD and EWC during FL training.
> 6. Section 4.3 (now 5.3) and the rest of the paper have been restructured to contain a clearer discussion of our alternative approach. We consider the observation that FFL is unhelpful in reducing the need to personalise models together with our alternative approach to be the main contribution.
>
> Please let us know if we have adequately addressed your concerns.
>
> [1]  Li, T., Hu, S., Beirami, A., & Smith, V. (2021, July). Ditto: Fair and robust federated learning through personalization. In International Conference on Machine Learning (pp. 6357-6368). PMLR.

---

### Official Review · Reviewer_C1c4 · 2022-10-28

**Confidence:** 3
**Correctness:** 3
**Technical Novelty And Significance:** 2
**Empirical Novelty And Significance:** 2
**Recommendation:** 5

**Clarity, Quality, Novelty And Reproducibility:**

Clarity - A well-written paper with useful background section.
Quality - Analyses appear competent and thorough.
Novelty - Novelty is hurt by the fact that the bulk of the work uses a previously published algorithm, just evaluated for a different purpose (suitability for personalization). The authors also present a novel and promising algorithm, but the work is preliminary.
Reproducibility - The authors do not open source their work, but the algorithm and datasets are open-source. The new approach is not open-source but may not be too difficult for a read to re-implement.

**Strength And Weaknesses:**

Strengths:
1. Presents detailed investigation of whether a fairness algorithm can also reduce the need for personalization, yielding the surprising finding that it does not reduce disparities between clients overall.
2. The background information is well written and provides a clear overview of fairness and federate learning.
3. Authors present results from a new algorithm based on distillation that can reduce the number of under-performing clients.

Weaknesses:
1. The main result is a negative result -- that Q-FFL did not reduce the need for personalization. This is unsurprising since Q-FFL is not a personalization algorithm. Since the authors present an alternative algorithm that does use fairness to reduce the need for personalization, it would be better if they led with this and used Q-FFL as a baseline.
2. Along the same lines, the results with their new approach seem too preliminary and under-developed. It is likely the authors will need to follow-up with another publication to present this approach in more detail. I strongly suggest to the authors to develop this work further and make it the main subject of their publication.
3. Color scheme in figure 3 makes it hard to distinguish red and orange dots, especially on such a small figure. Maybe use black for one of these symbols.

**Summary Of The Paper:**

This paper asks the question of whether the Q-Fair Federated Learning (Q-FFL) algorithm can help or obviate personalization. This federated learning algorithm attempts to instill fairness by weighting losses more heavily for clients with large losses, controlled via an exponential weighting parameter q. By equalizing the loss for under-performing clients, the authors hypothesize that Q-FFL may reduce the need to use personalization to boost individual client performance. However, they find through an offline study that Q-FFL does not reduce and can even increase the number of underperforming clients. They instead propose to use distillation to train local models that still benefit from global learnings. Preliminary results show this helps reduce the number of underperforming clients, indicating this as a promising new direction.

**Summary Of The Review:**

I am marginally uninclined to accept this paper. I believe there is a very promising piece of work here. The authors present a strong and clear background for the problem of fairness in federate search and explore an interesting question about whether fairness can reduce the need for personalization. This is an important topic, and the authors have made progress by showing one existing algorithm q-FFL does not meet this criteria, and they propose an alternative approach that performs better. However, the work is quite preliminary, meaning the authors focus most of the paper on the under-performing baseline. This paper would benefit by giving the authors more time to flesh out their new approach.

---

> ### Author Response · Authors · 2022-11-19
> **Author Responses to Reviewer C1c4**
>
> Thank you for your thoughtful comments. We have further expanded the overview of FFL and personalization works and have generalised our alternative approach from just distillation to allow for a broader class of loss functions. With regard to your concerns:
>
> 1. We do intend to explore further Multi-task Learning approaches for FL in the future, which would open with a proposed algorithm. However, this work was conceived as an experimental study on the interaction between fairness and personalization, with the secondary algorithm being introduced only when FFL failed to provide benefits. As such, while leading with a new approach may be a better choice, generally, it would require a complete re-framing of this paper, including its title, which could only be done with a new submission.
> 2. While Q-FFL is not, in itself, a personalization algorithm, a client is defined as needing personalization if the federated model offers a lower accuracy than one trained purely locally. Since Q-FFL can shift the emphasis on which the federated model underperforms, it should lead to fewer clients with very low accuracy. Thus, the clients for whom the federated model performs worst would be in less need of personalization. The paper finds that such benefits fail to materialise for the presented datasets because some clients represent easy training tasks and can train very high-performance local models. As such, raising the floor of the worst performers is not enough without also improving accuracy on clients for whom creating a good local model is trivial.
> 2. Thank you very much for your recommendation. For now we have generalised and further detailed our alternative approach in a manner which allows multiple personalization techniques to be used across rounds. Further experiments are provided testing the effectiveness of EWC instead of KD in reducing the need for personalization.
> 3. We have changed the colour scheme for the entire paper to make it easier to follow.
>
> If you have any further ideas for this paper or a future version of the publication please let us know so we may attempt to incorporate them.

---

### Decision · Program_Chairs · 2023-01-20

**Decision:**

Reject

**Justification For Why Not Higher Score:**

See major concerns described in (A) above.

**Justification For Why Not Lower Score:**

N/A.

**Metareview: Summary, Strengths And Weaknesses:**

The bulk of the contribution lies in verifying that the Q-Fair Federated Learning (Q-FFL) algorithm does not always help in providing a better starting point for personalization. The authors then follow up with knowledge distillation during FL training to improve the local accuracy.

While the reviewers have found this work interesting, they also voiced (A) several similar major concerns about this work and provided useful recommendations to this work. In particular, the reviewers think that a greater focus should be placed on their proposed work, the claim/findings should be investigated in a more general context, the qualitative and quantitative comparison with related work should be improved, a theoretical analysis would be useful, among other concerns.

The authors have made a major revision to their manuscript in response to the reviewers' feedback. However, we all agree that the revised version will need another thorough, careful review and hence, it is more appropriate to be resubmitted to a future ML-related venue.